# Recent Approaches in Magnetic Nanoparticle-Based Biosensors of miRNA Detection

Simge Balaban Hanoglu [1,*], Duygu Harmanci [2] , Nursima Ucar [1], Serap Evran [1] and Suna Timur [1,2,*]

1. Department of Biochemistry, Faculty of Science, Ege University, Bornova, Izmir 35100, Turkey
2. Central Research Test and Analysis Laboratory, Application and Research Center, Ege University, Bornova, Izmir 35100, Turkey
* Correspondence: simge93balaban@gmail.com (S.B.H.); suna.timur@ege.edu.tr (S.T.)

**Abstract:** In recent years, magnetic nanoparticles (MNPs) have been widely used in many fields due to their advantageous properties, such as biocompatibility, easy modifiability, and high chemical stability. One of these areas is the detection of cancer. It is essential to use existing biomarkers, such as microRNAs (miRNAs), for the early diagnosis of this disease. miRNAs are challenging to distinguish and detect in biological samples because they are small, circulating molecules. It is necessary to use more sensitive and feature-rich systems. Thanks to their large surface areas and magnetic moments, MNPs allow easy separation of miRNA at low concentrations from complex samples (urine and blood) and rapid and specific detection in biosensing systems. Here, we discussed the synthesis and characterization methods of MNPs, their stabilization, and MNP-based biosensors in terms of miRNA detection. We considered the challenges and prospects of these biosensor systems in evaluating the development stages, sensitivity, and selectivity.

**Keywords:** magnetic nanoparticles (MNPs); magnetic sensing; biosensors; cancer biomarker; microRNA (miRNA)





## 1. Introduction

Magnetic nanoparticles (MNPs) can be manipulated by a magnetic field [1]. They can generate responses in magnetism, and their small size is related to this response. They have many superior properties to large-scale materials with the same features because they have the chemical, mechanical, and magnetic capabilities that both nanomaterials and magnetism offer. These capabilities, which vary depending on their properties, have paved the way for them to attract attention and be used for various purposes in many fields [2].

MNPs can help isolate biomolecules that are difficult to separate from biological materials due to their complex matrices. They can also be used in water purification by increasing the sensitivity of existing systems, enabling the development of diagnostic support systems, and even improving and targeting treatments [3]. Especially in biomedical processes, they can immobilize biomolecules such as antibodies, proteins, enzymes, and DNA by binding to them, enabling their separation from complex mixtures with high efficiency. In this way, they can differentiate and determine biomolecules that can potentially be biomarkers but not easily detected in various biological samples [1,4]. One of these biomolecules is microRNA (miRNA).

After the Human Genome Project, findings on the non-coding part of the genome and the acceleration of studies in this field led to the discovery of miRNAs in 1993. miRNAs are small non-coding RNAs [5,6]. The evidence of their involvement in physiological and pathological processes is increasing day by day. Hence, their potential as biomarkers of various diseases indicates that we can also evaluate them as diagnostic molecules [7,8]. The main challenge is getting standardized and pure miRNA from biological samples, which could be obtained by non-invasive or minimally invasive methods [1,9]. The ability of MNPs to bind these nucleic acid fragments and release them in a reusable form can

be used for this purpose. The content of miRNA in these biological samples is relatively low [3,10,11]. It is possible to use the superior capabilities of MNPs to overcome the difficulties in isolating total miRNA, which is scarce anyway, from the existing sample and to detect miRNA with biomarker potential from this total miRNA. There are methods such as quantitative real-time polymerase chain reaction (qRT-PCR), microarray, northern blot, and modified invader test that we still use to detect miRNA today [12]. Besides these methods, biosensing systems are also being developed to detect specific miRNAs. MNPs can also be used in biosensor systems developed for miRNA detection. MNPs, along with other electrochemical, optical, plasmonic, and fluorescent sensing technologies, enable the development of systems for miRNA detection. Their adaptation to these systems is also related to the simple functionalization properties of MNPs. MNP-based sensing systems for miRNA detection are prevalent [1,4,13].

In this review, we have attempted to present the current literature from the last five years on magnetic nanoparticles, especially for detecting miRNAs. In addition to the properties and applications of magnetic nanoparticles, the biological significance of miRNAs and the applications mentioned above of magnetic nanoparticles in biosensing systems for miRNA detection were mentioned.

## 2. MNPs

One of the most critical nanomaterials is MNPs. They can be prepared on a large scale between 1 and 100 nm from pure metals such as iron, cobalt, nickel, or their metal oxides, as well as from mixtures of metals and polymers [14]. They consist of two parts: the core and the shell. The magnetic properties are associated with the core and shell, which is active in biomolecule recognition, binding, and catalytic processes [15]. Small particle sizes and a large surface area-to-volume ratio are the characteristics of MNPs. They are easy to synthesize and can be functionalized as desired during the production of the shell parts.

The motion of particles with mass and electric charge causes magnetism. MNPs exhibit superparamagnetic properties at high temperatures. Superparamagnetism occurs when the net magnetic dipoles are zero. A dipole is induced when an electromagnetic field is applied above a specific temperature. This situation causes the magnetic moments to align. However, when this field is removed, the magnetic moments become random again. The crystallinity and size of the structures, the type of the material, and the number of spins determine the superparamagnetism [16,17]. In addition, MNPs have a magneto-caloric effect defined as an adiabatic temperature change caused by the change in entropy of the material under a magnetic field [18]. Thanks to these exclusive properties, MNPs have a wide range of applications.

### 2.1. Synthesis

For MNP synthesis, different methods are used for the desired properties, such as size, morphological structure, compatibility, and stability. There are three main routes: chemical, physical and biological synthesis, and each method has advantages and disadvantages [19]. The methods used for MNP synthesis are summarized in Table 1.

**Table 1.** There are some studies about the methods used for MNP synthesis.

| Target MNP | Method | Advantage | Disadvantage | Ref. |
|---|---|---|---|---|
| Carbon-encapsulated MNP | Ball milling | • Easy method<br>• Suitable for large-scale production | • High energy<br>• Amorphous carbon encapsulation | [20] |
| $\varepsilon$-$Fe_2O_3$ nanoparticles | Ball milling | • Easy method<br>• Suitable for large-scale production | • Extend time for process | [21] |
| MNP | Laser ablation | • High purity<br>• Economic<br>• Short reaction time | • Wide-range particles<br>• Device requirement | [22] |

**Table 1.** *Cont.*

| Target MNP | Method | Advantage | Disadvantage | Ref. |
|---|---|---|---|---|
| $Fe_3O_4$ nanoparticle | Co-precipitation | • Easy method<br>• Short reaction time | • Depending on reaction condition | [23] |
| Zinc- and manganese-co-doped magnetic nanoparticles | Co-precipitation | • High reproducibility<br>• Simple reaction condition | • Requirement of stabilization agents<br>• Large particle size distribution | [24] |
| $FeCO_3$ | Thermal decomposition | • Simple, economical, environmental-friendly | • Dependence on reaction conditions | [25] |
| $CoFe_2O4$ | Thermal decomposition | • Simple method | • Dependence on reaction conditions | [26] |
| $Fe_3O_4$ | Hydrothermal method | • Economic<br>• Simple<br>• Scalable method | • Dependence on reaction conditions | [27] |
| Iron oxide nanoparticle | Hydrothermal method and biological synthesis | • Low-costly<br>• Eco-friendly | • Long procedure | [28] |
| Silica-Coated $Fe_3O_4$ Nanoparticles | Microemulsion | • Controllable size nanoparticles | • Using solvent | [29] |
| $Fe_3O_4$ | Sol-gel | • Well-crystallized, pure, spherical, and monodispersed MNP | • Dependence on reaction condition | [30] |
| $Mg_{0.5}Zn_{0.5}FeMnO_4$ magnetic nanoparticles | Green Sol-gel | • Eco-friendly | • Long procedure<br>• High temperature | [31] |

The two main approaches to physical synthesis are bottom-up and top-down. In bottom-up synthesis, large bulky materials are reduced to nanometer size, while in top-down synthesis, the atoms that form MNPs are combined into a nucleus and grow. The ball milling process is an example of top-down synthesis. Ball milling, developed by John Benjamin, is a mechanical synthesis method based on the dislocation of large materials by the impact of a ball and the fusion of the resulting particles to obtain magnetic particles in the nanostructure [32]. The production of carbon-coated MNP was achieved by ball milling in the study by Zhang et al. [20]. MNP was synthesized from $Fe(NO_3)_3$ and dopamine in stainless steel grinding tanks. The method provides MNP synthesis with an average diameter of ten nanometers and is helpful for industrial-scale production. However, it is not economical due to the high temperature and long period of 6 h required for production. Ball milling was used to synthesize $\varepsilon$-$Fe_2O_3$ nanoparticles [21]. The particles were obtained after milling at room temperature for 5 h, drying under pressure, and subsequent annealing for 4 h. $\varepsilon$-$Fe_2O_3$ with an average size of 15 nm was synthesized after this long processing time. The main drawback of the ball milling method is the large-scale particle size of MNP and contamination [33].

The laser ablation method is an example of a physical synthesis method. The solid raw material used for MNP synthesis is irradiated by laser light. Compared to ball milling, the size and shape of the particles can be controlled. As a disadvantage of the method, it is reported that, when the laser is used for a long time, the path of the laser is blocked by particles [34]. In the study by Svetlichni et al. [35], MNP was synthesized using two different laser ablation methods: water-pulse laser ablation and air-pulse laser ablation. It was found that the surface composition and electrokinetic properties were different. While it is said that the particles were spherical and dispersed in the range of 2–80 nm in the water-based method, particles with a larger scale of up to 2–120 nm and even 1 μm were obtained in the air-based method. In the other study [22], the laser ablation method synthesized MNPs from samarium-cobalt as a target material. The obtained MNPs had an evaluation of their antibacterial effect. Compared to the ball milling method, although it is possible to obtain MNP in a short time (30 min), MNPs were obtained in a wide range of 10–60 nm.

Biological synthesis has an advantage in the field of application due to its biocompatible properties [36]. Plants and living organisms (bacteria, viruses, fungi, etc.) are used for the synthesis of MNP (Figure 1). The mechanism of this method is not precise. The biological synthesis method is cleaner and environmentally friendly compared to chemical synthesis methods. It can be explained as chemical synthesis methods use toxic substances, while biological synthesis uses enzymes present in living organisms. This method obtains the raw materials for MNP synthesis from living organisms. Therefore, it is more challenging to obtain MNP with this method. It is necessary to clarify the mechanism of synthesis to control the size and shape. Moreover, it is necessary to overcome the problem of yield [37].

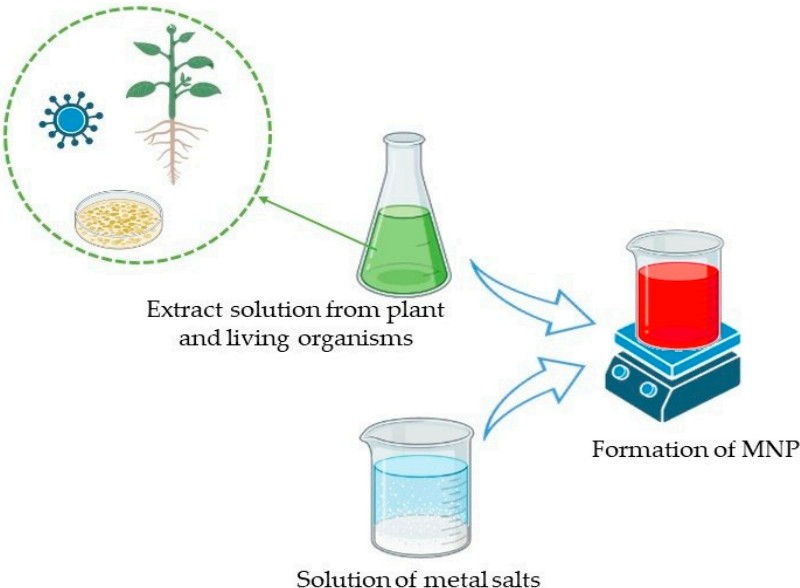

**Figure 1.** The biological method for the synthesis of MNP.

The chemical synthesis methods are based on bottom-up approaches. The preferred methods are co-precipitation, thermal decomposition, hydrothermal synthesis, microemulsion, and the sol-gel method [19]. Co-precipitation is a method based on mixing salt solutions of metal ions at room or high temperature, followed by the slow addition of a precipitant such as ammonia, hydroxide, or sodium carbonate solution to the solution to obtain MNP [38]. For example, the co-precipitation synthesis of $Fe_3O_4$ was carried out at room temperature using different bases (NaOH, KOH, or $(C_2H_5)_4NOH$) [39]. The reaction mechanism is explained as follows:

$$2FeCl_3 + FeCl_2 + 8BOH \rightarrow Fe_3O_4 \text{ (s)} + 4H_2O + 8BCl, \text{ (BOH: different bases)}$$

The synthesis method based on the mechanism of nucleation and growth is easy to implement. The disadvantage of the co-precipitation method is the difficulty in controlling the shape of the MNP. The size and composition of the MNP can be controlled by adjusting parameters such as pH, salt concentration, and temperature of the reaction medium [40]. In a study conducted to investigate the effect of differentiation of reaction conditions [23], MNP was synthesized by the co-precipitation method under two different conditions. The synthesis was done by changing the reaction temperature (25 and 80 °C) with or without adding nitrogen gas. While the size of MNP (MNP1) was 86.01 nm at 25 °C and $N_2$ gas environment, the size of MNP (MNP2) was 74.14 nm as a result of the reaction carried out at 80 °C without $N_2$ gas. It was also found that MNP2 was more stable than MNP1. The large particle size distribution and the difficulty of morphology control in co-precipitation synthesis were mentioned in another study [24]. For this reason, it is mentioned that the use of stabilizers is required as well as the synthesis of 10–15 nm $Fe_3O_4$ using polyethylene glycol (PEG), a cheap and biocompatible polymer.

Compared to co-precipitation, thermal decomposition, which is one of the most efficient chemical syntheses, allows the synthesis of monodisperse MNP with narrow size distribution and high crystallinity at higher temperatures [41]. The method uses organometallic precursors decomposed into organic surfactants at high temperatures. In addition, stabilizers such as fatty acid and oleic acid are used, which allow controlling the size and shape of MNP by slowing down the decomposition phase. The thermal decomposition of $Fe_3O_4$ nanocrystals at 245 °C in 2-pyrrolidone was discussed by Li et al. [42]. The formation mechanism is explained:

$$2 - \text{pyroIidon} \xrightarrow{\Delta} CO + \text{Azetidine}$$

$$\text{FeCl}_3 . 6H_2O \xrightarrow[\rightarrow]{\overset{\Delta}{\text{Azetidine}}} FeOOH + 3HCl + 4H_2O$$

$$6FeOOH + CO \xrightarrow{\Delta} 2Fe_3O_4 + CO + 3H_2$$

One disadvantage of the method is that the range of application of the obtained MNP is limited due to the use of organic solvents [43]. Moreover, the shape and size of MNP can be controlled by many parameters, such as the reaction time and the type of solvent and surfactant for the thermal decomposition method [44]. Wang et al. [25] studied the effect of reaction conditions on particle size through thermal decomposition. The size and shape of $FeCO_3$ were studied concerning the amount of oleic acid used and the reaction temperature. As the temperature increased, the dissolution of nuclei in the reaction medium decreased, and fewer were formed. The nanoparticles were more prominent due to the fact that there was more monomer in the medium. More stable metal complexes were formed when the amount of oleic acid increased. In this case, the activity of the intermediate decreased, and the nucleation rate and number decreased. In another thermal decomposition study [26], the structures of the particles obtained by increasing the synthesis temperature from 160 °C to 220 °C in the synthesis of cobalt ferrite nanoparticles are converted from rod shape to hexagonal shape. Furthermore, when the amount of ethylene glycol used as a surfactant is increased, a transformation from a hexagonal to an octahedral structure was observed.

The hydrolysis and oxidation reactions are carried out in high-pressure reactors or autoclaves in the hydrothermal method, which is another chemical synthesis method [45,46]. In this method, using aqueous or non-aqueous solutions, crystalline MNPs are formed, which are strongly dependent on the reaction time, the amount of pressure, and the temperature. $Fe_3O_4$ rod-shaped nanocrystals were synthesized using the hydrothermal synthesis method by Xi et al. [47]. The related reaction formation mechanism is explained as follows:

$$Fe^{2+} + 2OH^- \rightarrow Fe(OH)_2$$
$$\tfrac{1}{2} H_2O + Fe(OH)_2 + 12\,NO_3^- \rightarrow Fe(OH)_3 + 12NO_2$$

$$Fe(OH)_2 + 2\,Fe(OH)_3 \rightarrow Fe_3O_4 + 4H_2O$$
$$3Fe(OH)_2 + NO_3^- \rightarrow Fe_2O_4 + NO_2^- + 3\,H_2O$$

The advantage of this method can be sorted in that during the development of magnetic properties under high temperatures, evaporation is reduced for occurring under pressure, and particles with high crystallinity and similar size are obtained [48]. Hydrothermal synthesis is used as a simple and efficient method for $Fe_3O_4$ production [27]. The influence of the reaction conditions on the morphological structure is discussed. As a result of the reaction carried out at different temperatures, different structures such as cubes, octahedral structures, and hemispherical structures were obtained. The cytotoxicity of the obtained MNPs was studied, and it was found that the cytotoxicity changes depending on the morphological structure. Moreover, the need for unique materials for high pressure and high temperature can be considered a disadvantage. In recent years, with the definition of biological synthesis as eco-friendly, hydrothermal synthesis and biological synthesis

have been discussed in a study by Tovar et al. [28]. A hydrothermal MNP synthesis was performed with *Moringa oleifera* leaves and $FeCl_3 \cdot 6H_2O$ and the obtained MNPs were used to examine the growth parameters of the corn plant. However, it cannot be said to be economical and simple because the leaves are dried for 12 h as pretreatment and then autoclaved at 250 °C for 15 h during hydrothermal synthesis.

A thermodynamically stable and clear mixture of two immiscible liquids (water and oil) is formed in the microemulsion method. Here, water and oil stabilized the surface film property of the surfactants. The final size of MNPs depends on the concentration of the surfactant [49]. A solvent is required to extract the obtained MNPs. The disadvantage of this method is that it is not suitable for large-scale production due to the low yield of MNPs and the use of solvents [50]. $Fe_3O_4$ nanoparticles were synthesized by the water-in-oil microemulsion method [29]. The antimicrobial effect of MNPs coated with silica and controllable in size was investigated. However, a large amount of butan-1-ol/n-heptane mixture is noticeable. This situation can cause a problem for large industrial amounts of MNP production due to the requirement of butan-1-ol/n-heptane mixture.

The sol-gel method is composed of two reactions: hydrolysis and condensation. This method obtains a colloidal solution by dissolving the metal salt in water or solvents [51]. The high temperature in the reaction medium enhances the interaction between the particles. This leads to the removal of the solvents from the medium. MNPs form a gel after complete drying [52]. The disadvantage is that the reaction is affected by the temperature, pH, salt content, and solvent, while the advantage of the method is that the structure and size distribution of MNPs are simple [53]. For example, the effect of temperature changes on magnetic behavior was studied [30]. Pure, good crystal structure and one-dimensional $Fe_3O_4$ were synthesized, and the magnetic properties decreased with increasing temperature. This is an indication that the synthesis conditions should be adjusted depending on the application range of MNP. While MNP with high magnetic properties is suitable for removing heavy metals and dyes, its decreasing magnetic properties have been found unsuitable for this use. In recent years, the green sol-gel method has been used to avoid solvents in sol-gel synthesis. In one study [31], $Mg_{0.5}Zn_{0.5}FeMnO_4$ MNP, which can be used as a catalyst for the decolorization of RB21 dye, was prepared using sol-gel. The use of solvents was avoided by using tragacanth gum as a natural gel. Although the use of solvents can be avoided, the method is not economical due to the high temperature (4 h, 600 °C) requirements.

In summary, in chemical methods, it is possible to adjust the size and shape and to obtain nanoscale magnetic particles by adjusting the reaction conditions compared to physical methods. Although the synthesis is time-consuming and the mechanism is not yet fully elucidated compared to physical and chemical synthesis, biological synthesis is promising due to its eco-friendly, reproducibility, and high efficiency. It cannot be concluded that there is only one correct method. Particle size, shape, yield, and cost are some of the parameters that should be considered when selecting the appropriate method.

### 2.2. Coating/Stabilization and Functionalization Strategies

As mentioned above, MNPs can be synthesized from pure metals such as iron, nickel, etc., and their alloys by various synthesis methods. One of the main challenges of MNPs synthesis is pure metals or their alloys' instability to oxidation. In addition, decreased diameter is related to this instability. The stabilization of MNP is essential for the development of strategies. The basis of these strategies is to consider the synthesized MNPs as core and to provide them with a shell to protect them from environmental factors [45,50].

Generally, suitable surfactants, polymers, inorganic metals, non-metals or oxides, or lipid structures are used as coating processes. Surfactants and polymers are some of the most common stabilizers used to prevent the aggregation of MNPs by increasing electrostatic repulsion. They are chemically or physically placed as a layer over the MNP cores to balance the magnetic attraction and van Der Waals forces between the particles [54]. Commonly, surface stabilizers with carboxylic acid, phosphate, sulfate groups, and polymer-based stabilizers

(PEG, polyvinyl alcohol (PVA), polyacrylic acid (PAA), starch, chitosan, etc.) can be used [55]. Here are some parameters to consider when stabilizing with surfactants and polymers. The crystal structure, molecular weight, and conformation of MNPs can be adjusted with the help of the monomer ratio structure in a polymer coating. The metallic magnetic nanoparticles coated with polymers or surfactants are unstable in air and in acidic solutions. Among their disadvantages is that the polymer coating is not temperature stable [56].

The core of MNP can be coated with various metals such as gold, cobalt, etc. They are suitable for coating due to their low reactivity and stability in the air. Moreover, especially in the case of gold, they facilitate the binding of MNPs to thiolate compounds with different groups (carboxyl, amino, biotin, etc.) due to their natural reactivity towards -SH groups [57]. One of the shells developed to protect MNP cores is silica. Silica is hydrophilic, and it facilitates modification. The contact of the core with other groups is protected by this shell [58]. However, it is not a suitable coating for primary pH conditions because it is unstable under alkaline conditions. Another coating material is carbon-based materials, such as graphene, which attracts attention due to its biocompatibility and stability. Carbon-coated MNPs have been reported to have higher magnetic moments than oxides [59]. Coating MNPs with lipid structures such as liposomes is one of the stabilization options, primarily to facilitate their use in application areas such as drug distribution and imaging [60].

### 2.3. Characterization

Characterization studies are critical to determining the accuracy of MNP synthesis and very critical properties, such as size, surface modification, composition, structural configuration, and magnetism. All characterization techniques of MNP are summarized in Figure 2.

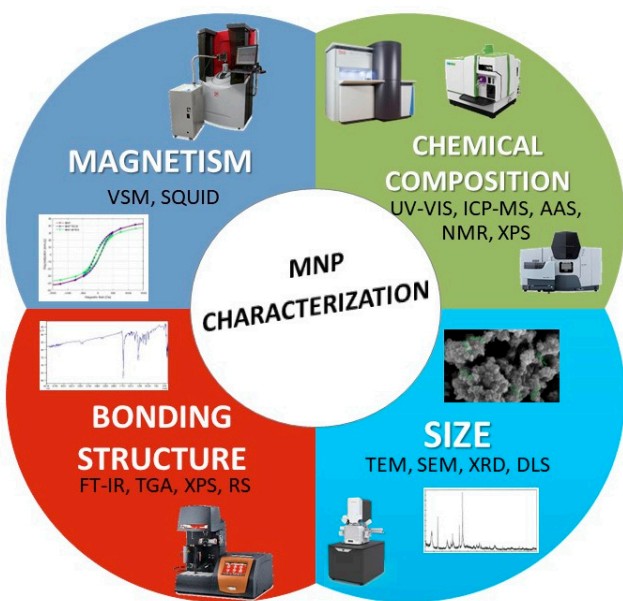

**Figure 2.** Characterization techniques for MNP.

Nowadays, transmission electron microscopy (TEM) and scanning electron microscopy (SEM) are most commonly used to determine the size and surface morphology of MNPs [61]. These techniques can provide information about their shape and size. In particular, the ratio between the core and shell size of MNPs are critical parameters as they affect the magnetic properties and behavior. SEM only provides information about the external structure of the particles, while TEM is a widely used method to study the core structure [62]. Due to the aggregation tendency of MNPs, sample preparation is critical for TEM and SEM. These methods are suitable for working with dry samples and are operator-dependent.

For this reason, these methods are quite open to personal mistakes as a disadvantage. Secondary methods are used to overcome these disadvantages. High-resolution TEM [63] and cryogenic TEM [64], which can analyze in aqueous or biological media, are examples. In addition, dynamic light scattering (DLS) is used for information on size, dispersion, and surface area [19]. DLS is one of the most popular methods for measuring colloidal dispersions because it is simple and easy to perform. In this method, the scattered light waves are measured as a function of a particle's hydrodynamic radius and Brownian motion. The measurement provides information on stability, interparticle interactions, aggregation and surface modification, and size measurement [65]. Zeta potential measurement, based on the electrostatic repulsion between particles from the charge potential of the particles, is also performed with DLS to determine particle stability. The parameters for DLS measurement are the MNP concentration, the solution chosen for particle distribution, and the pH. These parameters are crucial for an accurate measurement, especially for the measurement of zeta potential, since the surface charge and the suspended state of the particles are very important [66]. X-ray diffraction (XRD) can provide information about crystal structure. It affects the magnetic property of the crystal structure of the atoms in the core structure. Therefore, it is crucial to know the crystal structure, size, and lattice spacing. This technique, used for powdered samples, is unsuitable for amorphous materials and particles smaller than 3 nm because of the wide of obtained peaks [67].

Chemical characterization, such as elemental mapping, is essential for understanding the interactions of MNPs with each other and the environment. For elemental analysis of MNPs, UV-Vis spectroscopy, mass spectroscopy (MS), inductively coupled plasma mass spectroscopy (ICP-MS), atomic absorption spectrophotometry (AAS), and X-ray photoelectron spectroscopy (XPS) are common techniques. UV-Vis spectroscopy can determine the concentration of elements contained in MNPs and characterize their various conjugates [68]. However, MS is more appropriate for analyzing MNPs with lower elemental concentrations. For elemental analysis of MNPs, ICP-MS and AAS containing more than one element are also used. With a lower detection limit than AAS, ICP-MS enables the characterization of the elemental with high selectivity and sensitivity [67]. On the other hand, XPS is one of the common methods for the chemical analysis of surfaces. In general, it provides information about elemental composition, the electronic structure of elements, and oxidation. In addition, this method can be used to analyze the core and shell structure and obtain information about surface functionality. The advantage of XPS is that the sample is not damaged. The disadvantage is that a solid sample is needed, making the result difficult to interpret [67].

In addition to XPS, Fourier transform infrared spectroscopy (FT-IR), X-ray absorption spectroscopy (XAS), thermogravimetric analysis (TGA), and Raman spectroscopy (RS) are used to characterize the binding structures in MNPs [19]. FT-IR is used to identify the binding energy, functional groups, and oxidation states of the bound structures. The XAS method, on the other hand, provides information about the electronic configuration in addition to the oxidation state [45]. TGA is used to elucidate the formation of surfactants and polymer structures used in the coating as well as the binding activity on the MNP surface by providing information about the mass of the particles [69].

Magnetism, the most remarkable property of MNPs, is one of the most critical parameters for characterization. Vibrating sample magnetometer (VSM) and superconducting quantum interference device magnetometry (SQUID) measurements are used to determine this feature [70]. The VSM method is a simple and inexpensive measurement of a magnetic moment that is voltage-dependent and uses temperature, field, and crystal direction as functions [71]. The SQUID measurement system allows the measurement of magnetism in samples in powder, liquid, gas, thin film, and crystal forms. This method, which is more sensitive than VSM, provides information about the magnetic properties of the material [66].

In summary, there are many methods for characterization, but more than a single method is required. Therefore, it is necessary to synthesize MNPs by focusing on their applications and desired properties and then conduct characterization studies using different methods simultaneously.

### 2.4. Application Area

MNPs are multifunctional and have magnetic and plasmonic properties, and these features have given rise to a wide range of applications for MNPs. Some of these applications of MNPs have been summarized in Figure 3.

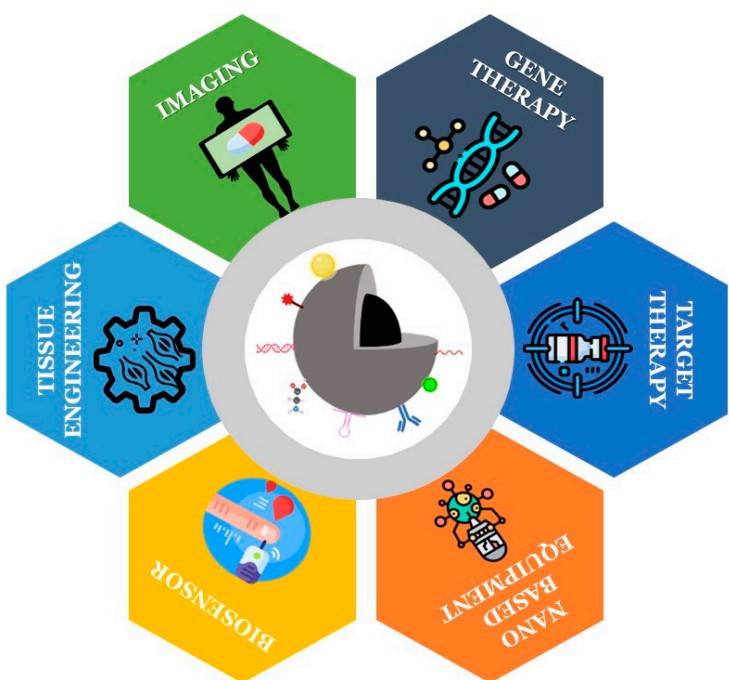

**Figure 3.** Application area for MNP.

Magnetic resonance imaging (MRI) uses MNPs as contrast agents and allows 3D imaging of cells or organs. Conventional contrast agents (gadolinium, etc.) have some problems, such as rapid elimination, low sensitivity, and specificity [72]. They are a potential candidate instead of contrast agents due to the strong magnetization, easy functionalization, and biocompatibility of MNPs. In the MR images, the molecules' relaxation times in the transverse and longitudinal directions are important; accordingly, tissues and organs are distinguished. Due to the magnetic moments of MNPs, they shorten this time and provide brighter and sharper images [73,74]. MNPs are also used in gene therapy with the magnetofection method. The desired gene is bound to the MNP surface thanks to its simple functionalization. They are delivered to the site determined as a target due to their magnetic properties. This process is called magnetofection and refers to magnetic field-based gene transfer. After the gene bound to the MNPs is transferred to the appropriate site by the external field, it is released into the environment using appropriate methods [75]. Targeted cell killing with specific drug delivery is another approach to targeted therapy using MNPs. Chemotherapeutics and radiotherapeutics are the primary tools used for targeted drug distribution. The basis of these treatments is to destroy cancer cells without harming healthy cells. Thanks to the conjugation property of MNPs, functionalization with biocompatible materials and the desired drug design can be achieved. In addition, MNPs functionalized with drugs can destroy cancer cells, not healthy ones. MNPs functionalized with drugs can destroy cancer cells by the effects of magnetic field and temperature due to their hyperthermia property [76–78]. In contrast to targeted cell death, the use of MNPs is very promising in the regeneration of damaged tissue or organs. Due to their large surface area and physicochemical properties, they can be used in stem cell therapy. With MNPs that can be manipulated with a magnetic field gradient, it is possible to direct cells to a region under a magnetic field, provide for tissue formation, and control their functions. Therefore, the use of MNPs in tissue engineering is on the agenda [79,80].

Recently, MNPs have been used to develop nano-based types of equipment, such as face masks and disinfectants, to protect from the COVID-19 pandemic [81]. In addition to this equipment, the COVID-19 pandemic has also brought to light the lack of rapid diagnostic systems at the bedside, called point-of-care (POC). One of the advantages of biosensors is that they are designed as point-of-care systems [82]. They give specific and sensitive measurements. MNP is used in the surface functionalization of biosensor systems [16]. Due to its sensitivity to magnets, surface modification with MNP can be performed quickly and cheaply. MNP-based biosensor systems are attracting attention not only for the detection of COVID-19 but also for many other areas, such as the detection of pathogens, illicit drugs, pesticides, and cancer [83–86].

## 3. MNPs for Detection of miRNA

### 3.1. miRNA

miRNAs are non-coding RNAs of 18–24 nucleotides in length and play essential roles in gene expression. miRNA biogenesis mainly follows the canonical biogenesis pathway [87]. Briefly, miRNA is transcribed by RNA polymerase II, and a microprocessor complex processes the primary transcript (pri-miRNAs). The resulting precursor-miRNA (pre-miRNA) is exported to the cytoplasm, where the ribonuclease Dicer processes it to generate an miRNA duplex. The miRNA duplex is then unwound, and the RNA strand that remains on the AGO protein acts as the mature miRNA [88].

Since their first discovery [89] thousands of miRNAs have been identified in different organisms, ranging from viruses [90] to fungi [91], and to higher eukaryotes [92,93]. With the advances in next-generation sequencing technologies, novel miRNAs could be identified [94,95].

miRNAs were found to be linked to several biological mechanisms, such as thermal stress response [96], regulation of immune cells in sepsis [97], atherosclerosis [98], and B cell receptor signaling [99]. The expression levels of miRNAs are usually changed in several pathological conditions such as multiple myeloma [100], non-small cell lung cancer [101], leukemia [102], Parkinson's disease [103], and diabetes-induced cardiomyopathy [104]. For instance, miRNA-937 was found to be overexpressed in colon cancer [105] and hepatocellular carcinoma [106]. The decrease in the level of miRNA-149-3p was shown to be associated with poor prognosis in oral squamous cell carcinoma [107]. Considering the correlation between changes in miRNA levels and diseases, miRNAs hold great promise as novel biomarkers. Thereby, miRNAs offer opportunities for early detection [108–110]. In particular, circulating miRNAs in biological fluids are significant for non-invasive diagnosis. The intracellularly produced miRNAs were found to be released in the extracellular environment through incorporation into the exosomes or by forming a complex with proteins [111,112]. Moreover, miRNA content in the extracellular vesicles was found to be affected by the disease state [113]. Thus, it was proposed that the miRNA derived from the extracellular vesicles could be used as a biomarker [114,115]. Interestingly, miRNA was found to be more stable than messenger RNA (mRNA) [116,117]. As stable biomarkers, miRNAs were detected in different types of extracellular fluids, such as urine [118], plasma [119], serum [120], cerebrospinal fluid [121], and saliva [122]. As a non-invasive approach, miRNA-based liquid biopsy was proven to be successful at detecting melanoma [123] and urothelial carcinoma of the bladder [124]. Given that the disease regulation mechanisms of miRNAs continue to be discovered [125–129], miRNAs attract much interest as non-invasive diagnostic and prognostic biomarkers.

### 3.2. Traditional Methods for miRNA Detection

Specificity and sensitivity are the two critical parameters for miRNA detection. Sequence similarity among miRNAs [130], and the need for detection at very low concentrations [131] are the challenging issues that should be considered. Northern blotting, RT-qPCR, and microarrays are the conventional methods with distinct advantages and disadvantages in terms of those two parameters [10,101,132].

### 3.2.1. Northern Blotting

Northern blotting, which is also known as Northern hybridization, is a classical technique to analyze RNA molecules [133,134]. RNA is first subjected to denaturing gel electrophoresis and then transferred onto a positively charged nylon membrane. RNA is fixed on the membrane by UV-mediated cross-linking. Then, it can be visualized after hybridization with a probe, which could be either radioactive or non-radioactive. Northern blotting is advantageous due to the fact that it is simple and reliable due to the sample being directly used after the isolation step. It also allows discrimination between miRNAs and their precursors [135]. However, it is time-consuming and not suitable for the analysis of different miRNAs having the same molecular weight. Low sensitivity and the requirement for large amounts of RNA are the other disadvantages.

Regarding miRNA analysis, several modifications to the original method were adopted to overcome those limitations. As an alternative to UV, 1-ethyl-3-(3-dimethylaminopropyl) carbodiimide hydrochloride (EDC) was proposed to cross-link RNA to the nylon membrane, which resulted in improved sensitivity [136]. Locked nucleic acid (LNA) modified oligonucleotide probes were successfully used against the sensitivity and specificity problems of DNA oligonucleotides [137]. In order to avoid the safety problems of radioactively labeled probes, digoxigenin (DIG)-labeled oligonucleotide probes containing locked nucleic acids [138], and biotin-labeled probes [138] were proposed. The protocol was also optimized for the detection of viral miRNAs [139]. In another study, liquid hybridization consisting of pre-hybridization of target RNA with oligonucleotide probes, Exo-1 digestion, and non-denaturing gel electrophoresis achieved greater sensitivity than Northern blot [140].

### 3.2.2. RT-qPCR

RT-qPCR is a gold standard for miRNA detection, enabling specific, sensitive, and quantitative results. For RT-qPCR analysis, target miRNAs are reverse-transcribed to complementary DNA (cDNA), which is then amplified by qPCR. Using intercalating dye or hydrolysis-based probes allows real-time fluorescence detection of the amplified products. Quantification is then achieved by using the relationship between the threshold value and the starting copy number of miRNA [141].

Although RT-qPCR is a well-established method, primer design and reaction conditions may need optimization for every new miRNA analysis [142,143]. Regarding the primer design issue in miRNA analysis, the tools such as miRprimer [144] and miPrimer [145] were proposed. A modified form of RT-qPCR was named the stem-loop RT-qPCR, which consisted of two steps [146]. The stem-loop primer was first hybridized in this technique to the miRNA molecule. Then, it was reverse-transcribed and used as a template in conventional qPCR, including the fluorescent-based probe. In a further study [147], a universal stem-loop primer was designed and shown to save 75% of the cost of primers and 60% of the test time compared to the stem-loop primer. Later, a universal hairpin primer system was proposed to eliminate the need for designing miRNA-specific hairpin primers, thereby reducing the cost [148]. RT-qPCR was used for the analysis of miRNA profile in embryonic stem cell differentiation [149] and detection of extracellular vesicle miRNAs [150], as well as to distinguish miRNA editing isoforms [151]. As RT-qPCR is sensitive and specific, it is still widely used for miRNA detection despite the challenges with optimization. In recent years, compared to conventional RT-qPCR, droplet digital PCR [152–154] has also been popular due to its better sensitivity and diagnostic potential.

### 3.2.3. Microarrays

Microarray technology relies on hybridization between the oligonucleotide probe immobilized on a solid surface and the target miRNA. Before hybridization, the target miRNA is reverse-transcribed to cDNA and simultaneously labeled. Then, the arrays are scanned for the hybridization signal, depending on the labeling strategy. Labeling can be achieved by the incorporation of fluorescence dyes [155,156] or radioisotopes [157]. Alternatively, miRNAs can be directly labeled with biotin, and the hybridization can be monitored by

a fluorescent signal that results from the binding between biotin-labeled miRNAs and streptavidin-conjugated quantum dots [158]. The protocols differ in terms of probe design, probe immobilization strategy, sample labeling, and signal detection [159]. For instance, instead of oligonucleotides, peptide nucleic acids (PNA) can be used as probes [160]. Furthermore, miRNA microarray data analysis [161] and data submission [162] issues need to be considered.

The major advantage of microarrays is that they enable a high-throughput screen for comparing miRNA expression profiles in different organs or tissues. For instance, miRNA microarray profiling allowed the identification of miRNAs that could be associated with liver cancer [163], attention deficit hyperactivity disorder in children [164], epithelial ovarian cancer [165], and rheumatoid arthritis [166]. Although miRNA microarray analysis is very useful for comparing miRNA expression levels between two states (e.g., control tissue and cancer tissue), it suffers from low sensitivity and low specificity. Similar to Northern blot analysis, it is not a suitable method for the discovery of novel miRNAs.

### 3.3. MNP-Based Biosensors for miRNA Detection

As described above, there are many conventional methods for detecting miRNA. These complex techniques require specially trained personnel and can give false positive results [10]. Therefore, new methods are needed to detect abnormal miRNA expression, which is critical for early diagnosis. With this approach, biosensor studies based on nanomaterials can be an alternative to conventional analytical devices due to their small sample size, low cost, fast response time, and ease of use. Biosensors provide measurable signals depending on the concentration of the target analyte. They are defined as analytical devices consisting of two main parts, a detection element and a transducer [57]. According to the type of signal generation (electrochemical, optical, thermal, etc.), it is possible to classify biosensors [167]. In recent years, MNP-based biosensor systems have been used for the detection of miRNA. The systems developed in the last five years are summarized in Table S1.

### 3.3.1. Optical Biosensor Systems

There are advantages to using optical biosensor systems in terms of noise-free, stable, and sensitive properties compared to other biosensor systems [168]. These biosensors are a good alternative for the detection of cancer markers, as they provide a non-invasive approach [169]. The optical sensing systems based on different signal conversion principles such as colorimetry, fluorescence, chemo/bioluminescence, and scattering-based biosensing, can be used to detect microRNA from biological samples such as plasma, serum, and blood [170].

The colorimetric optical sensor is an analytical system that measures the emitted or absorbed light intensity resulting from the recognition of the target molecule by the bioreceptor [171]. The biosensing that occurs in these systems is converted into a color change. Nanomaterials such as magnetic nanoparticles and gold nanoparticles have been widely used for this purpose. These systems can be described as simple, practical, and cheap because they can be read visually and without any tools [172]. In MNP-based colorimetric biosensor systems developed for miRNA detection, hemin chemistry [173], colorimetric TMB-$H_2O_2$ systems [174], and aggregation of gold nanoparticles [175] in salt were used and the color changes were monitored with UV-Vis absorption spectrum. Colorimetric sensor platforms were achieved by converting a colorless substrate to a green product in the plate for the detection of miR-21 (Figure 4a) [173]. Gold decorated MNPs (GMNP) were used in the sensor platform based on the catalytic hairpin assembly (CHA) reaction. In the presence of the cofactor hemin, oligonucleotides were labeled with GMNP to form a colored product. As a result of the reaction in the presence of $H_2O_2$, the LOD was determined to be 1 aM and the reaction time was reduced to less than four hours. Another colorimetric sensor system was developed for the detection of Lethal-7 (let-7a) miRNAs in gastric cancer [174]. In this system, $Fe_3O_4$ nanosheets were functionalized

with target miRNA (let-7a), and hairpins H1 and H2. Thanks to the hybridization chain reaction, a double-stranded DNA (dsDNA) structure was formed on the nanosheet. After removing the excess $Fe_3O_4$/dsDNAs in the medium with a magnet, $H_2O_2$ and TMB were added. LOD was determined to be 13 aM as a result of the isothermal process without enzyme. In the colorimetric sensing system developed by Wang et al. [175], the ability of MNPs to isolate the target analyte from the sample with a single magnet was exploited. Subsequently, the miRNA-155 functionalized with AuNP was precipitated by the salt-induced aggregation method. A two-part colorimetric sensor system was developed. First, $Fe_3O_4$ nanoparticles were coated with gold, then functionalized with DNAzyme. Single-stranded DNAs (ssDNAs) were formed in the presence of miRNA-155. ssDNAs were separated with a magnet and quantified by precipitation with NaCl in AuNP. It was possible to detect miRNA within 2 h at room temperature using the LOD in the fM level.

Fluorescence-based optical sensors are widely used for the detection of small biomarkers such as DNA due to their advantageous properties, such as low cost, high efficiency, and easy processing steps. In these platforms, the light emitted from the target sample is measured as a result of the excitation. In fluorescence-based biosensor systems developed with MNPs for miRNA detection, dyes (fluorescein (FAM) [176–178]) and substrate (tyramine) [179] are used for fluorescent labeling. Measurements are simplified with fluorescence resonance energy transfer (FRET) [178] or quenching [180]. Two different fluorescence systems have been developed for the potential prostate cancer biomarker miRNA-141. One study by Sun et al. [176] was based on duplex-specific nuclease (DSN), while the other study by Wang et al. [179] was based on the toehold-mediated strand displacement reaction (TSDR) using the horseradish peroxidase (HRP) enzyme. In the study by Sun et al., $Fe_3O_4$ was used as the MNP, and the surfaces of the MNPs were coated with polydopamine [176]. In the presence of the target miRNA, it hybridized with 6-carboxy fluorescence (FAM)-labeled single-stranded DNAs. Due to duplex-specific nuclease (DSN) in the medium, the FAM-labeled DNA was separated from the RNA and fragmented into small fragments. When the remaining DNA was absorbed on the $Ca^{2+}$ surface of the MNPs, a new cycle began in which the miRNA hybridized with a new DNA. The released small particles FAM-DNA caused strong fluorescence intensity, and the miRNA could be quantified based on the changes in fluorescence. For this method, LOD was determined as 0.42 pM. It is quite remarkable that it is used for the detection of miRNA from human cell lines. According to the study by Wang et al. [179], the LOD of exosomal miRNA-141 could be reduced to fM by detecting it with MNP-based HRP-catalyzed TSDR (Figure 4b). For this purpose, 4 different probes were immobilized on the surface of carboxyl terminated MNPs. The miRNA-141 sequence recognized the toehold domain complementary part of the probe1, and as a result of the TSDR reaction, P2 was removed from the environment while a new toehold part was formed. Upon the addition of biotin-tipped probe 4, probe 4 and probe 1 formed a double chain. miRNA-141 released thus initiated a new cyclic amplification. After the addition of streptavidin-HRP (SA-HRP), MNP was collected from the medium with a magnet. Then, tyramine, one of the fluorescent substrates, was added to the MNP. The color change that occurred under $H_2O_2$ catalysis was used to calculate the miRNA concentration. In addition, a fluorescence biosensor using MNP and Clustered Regularly Spaced Short Palindromic Repeats (CRISPR)/CRISPR-associated proteins system family (Cas12a) was developed for the first time to detect exosomal miRNA [180]. In this study, Cas12 increased target specificity because it could bind ssDNA or double-stranded DNA (dsDNA) without the contiguous protospacer motif, and thanks to MNPs, the trans-splicing activity of CRISPR/Cas12a was abolished in the absence of the target miRNA. This is a remarkable study in terms of the detectability of exosomal miR-21 from lung cancer plasma with high sensitivity and specificity.

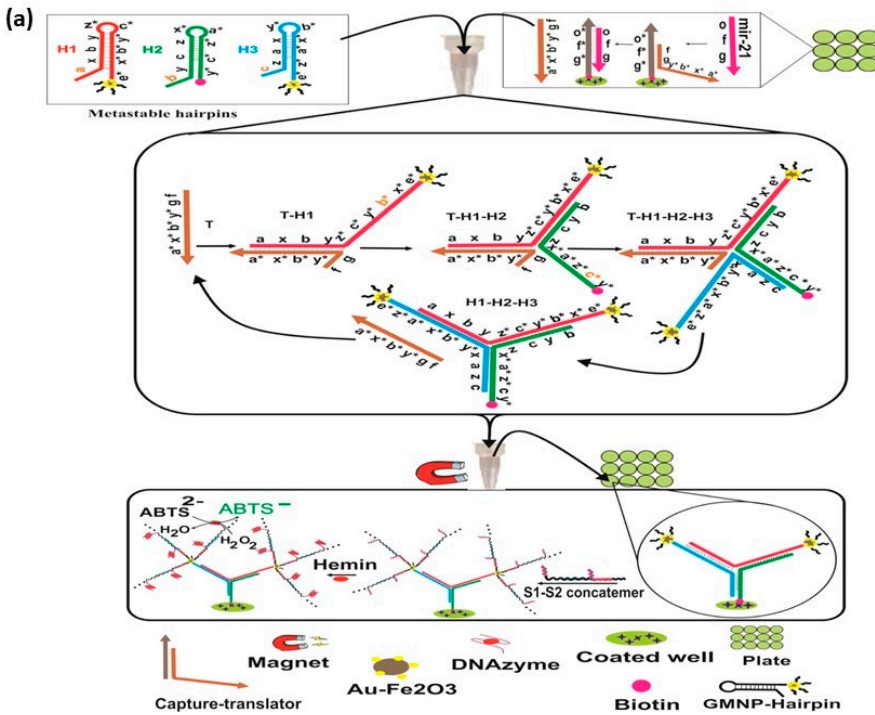

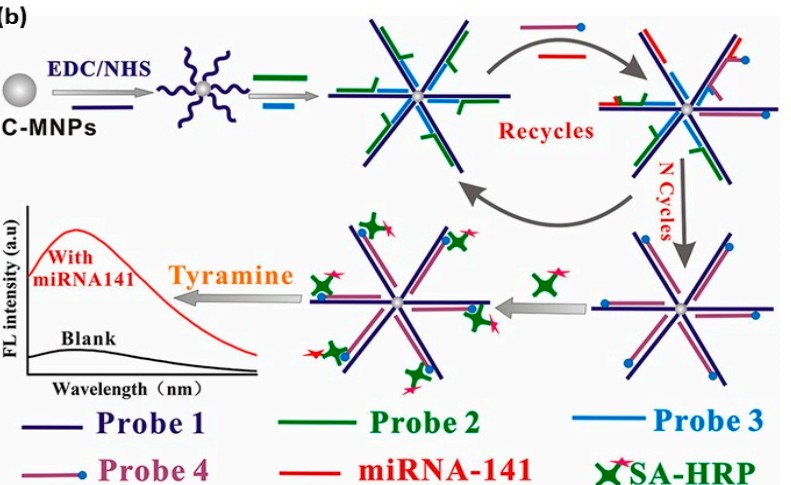

**Figure 4.** MNP-based (**a**) Colorimetric-based optical biosensor platform. H1 and H3 are GMNP-labeled, while H2 is biotinylated. The segments that are exposed and involved in the serial reactions required for the formation of the targeted nanostructure are shown with "*". [Reproduced with the permission of [173] Copyright 2019 Published by Elsevier B.V.]; (**b**) Fluorescence-based platform optical biosensor platform [Reproduced with the permission of [179] Copyright 2019 Published by Elsevier B.V.].

Electrogenerated chemiluminescence sensing is a detection method that originated from a combination of electrochemical and chemiluminescent sensors. Basically, an electron transfer reaction takes place on the electrode surface [181]. This strategy could be used for clinical detection of miRNA [182–184] because of its rapid response, high sensitivity, and low cost. Wang et al. [182] designed a multichannel paper-based electrochemiluminescence microfluidic platform that focused on the detection of two different miRNAs 155 and 126. After subjecting the paper to a wax printing process to enable the fabrication of bipolar electrodes, the electrode sites were prepared with AuNP. Two different surfaces were designed for miRNA detection. Signal probes with CdTe quantum dots (QD) and Au@g-

$C_3N_4$ nanosheets were used for the detection of miRNA-155 and miRNA-126, respectively. In this system, K2S2O8 was used as a co-reactant, while $Fe_3O_4$ MNP was assigned as a carrier. It is important to note that the luminescent light on both platforms is stronger than the single electroluminescent signal. In addition, the use of $K_2S_2O_8$ was found to improve the selectivity, sensitivity, response speed, and signal intensity of the sensor. The use of nanosheets in electrochemiluminescence systems also attracts attention. These structures were used in a sensor system for miRNA-210, a breast cancer marker. Due to their optoelectronic properties, ease of synthesis, biocompatibility, and ability to absorb ssDNA structures, analysis from serum was achieved [183]. In addition, $SiO_2$-coated $Fe_3O_4$-NPs were functionalized with cholesterol-linked aptamers and were immobilized on the surface of the magnetic carbon electrode. The $MoS_2$ nanosheet-DNA probe was attached to the surface. The LOD value was determined as 0.3 fM using the luminescence change.

Another interesting system in the development of MNP-based optical platforms is the surface-enhanced Raman spectroscopy (SERS) based sensor system. This is a surface-sensitive technique used for specific detection that measures the interaction between nanostructures and light by Raman scattering [185]. Recently, the method has also been used for the detection of miRNA [186–189]. SERS tags are used for optical measurement. With these tags, a single target can be analyzed in a single system, while it can be prepared in multiplexed systems. By Zhang et al., two different systems have been developed for miRNA-141 alone [189] as well as a multiplex system for simultaneous analysis of miRNA-141, -429, and -200b [188] sing the same SERS-nanotag and its substrate. In both systems, silica-coated, analyte-embedded Au nanoparticles (SA@GNPs) and Au-coated paramagnetic nanoparticles (Au@MNPs) were used as SERS-nanotag and substrate, respectively. For the detection of miRNA-141 alone, 5,5′-dithiobis(succinimidyl-2-nitrobenzoate) (DSNB) was used as SERS-tag, whereas DSNB, methylene blue, and Nile blue were used as SERS-tag for the determination of miRNA-141, -429, and -200b in the multiplex system, respectively.

In addition to all these optical sensors, optomagnetic sensor systems were developed for the detection of miRNA. These systems consist of a photodetector, a light source, and a magnetic actuator to generate a magnetic field. It is a system that has become a trend in recent years due to advantages such as the suppression of noise, low cost, and the increase in the reaction rate to the molecules colliding with the action of the magnetic force [190]. Several optomagnetic sensing systems have been developed for let-7b miRNA, a member of the let7 family [191,192]. In both systems, MNPs released as a result of the recognition of the target miRNA by the designed surface could be measured by a laser-based optomagnetic sensor. By analyzing serum in both systems, it offers potential for use in clinical applications.

### 3.3.2. Electrochemical Biosensor Systems

Electrochemical systems are one of the most commonly used sensors for the detection of miRNA. A reaction between the target analyte and the biological recognition element is converted into an electrical signal in these systems. Since this reaction and the flow of electrodes occur in the electrode system, the electrode is one of the essential components in these systems [169]. The counter, reference, and working electrodes are used. Generally, Ag/AgCl is used as the reference electrode, and Pt is used as the counter electrode. The working electrode can be made of materials such as gold, carbon, or graphene and can be used specifically [193]. Electrochemical platforms categorized according to the measurement principles of amperometric, potentiometric, voltammetric, and impedimetric are in great demand due to their advantageous properties such as ease of operation, miniaturization and fabrication, and low cost [194].

Voltammetric systems based on measuring the change in current with a change in potential are common electrochemical sensors. These techniques in various forms, such as chronoamperometry [91], cyclic voltammetry (CV) [195,196], square wave voltammetry (SWV) [197–199], and differential pulse voltammetry (DPV) [196,200–202], are also commonly used for miRNA detection due to their sensitivity and wide linear range. In addition,

electrochemical biosensors are functionalized or combined with MNPs because they are easy to functionalize, inexpensive, and have the ability to specifically detect target miRNAs from the sample under the magnetic field. In this regard, MNPs can also play a role in the sensing system by capturing miRNAs from body fluids such as serum and plasma and simply connecting them to the electrode surface using a magnet. By Shen et al. [197], MNPs were functionalized with gold nanoparticles and gold stir bars in the electrochemical sensing system designed for the simultaneous detection of miRNA-21 and miRNA-155 (Figure 5). They were both used to capture miRNAs from serum samples and electrochemical measurements. In this system, $Fe_3O_4$ nanoparticles were functionalized with electrochemical labeled and complementary DNAs after gold plating and immobilized on the SPCE surface with magnets. SWN measurements were performed on target miRNAs captured by DNA/RNA hybridization. The method has the advantage of being easy to use and increasing amplification efficiency, not requiring enzymes. Using a similar strategy, another electrochemical sensor system was developed by Tavallay et al. [198]. It is also reported to be the first system that can detect miRNA from unprocessed blood samples. For the analysis of miR-21, gold-coated magnetic nanoparticles were used and immobilized on the surface of the gold microelectrode using a magnet. Direct analysis of miR-21 in untreated blood from a lung cancer mouse model was achieved. The sensitivity was reported to be better than other sensors with a LOD of 10 aM. It was also much faster than conventional PCR, providing the analysis result in less than 30 min.

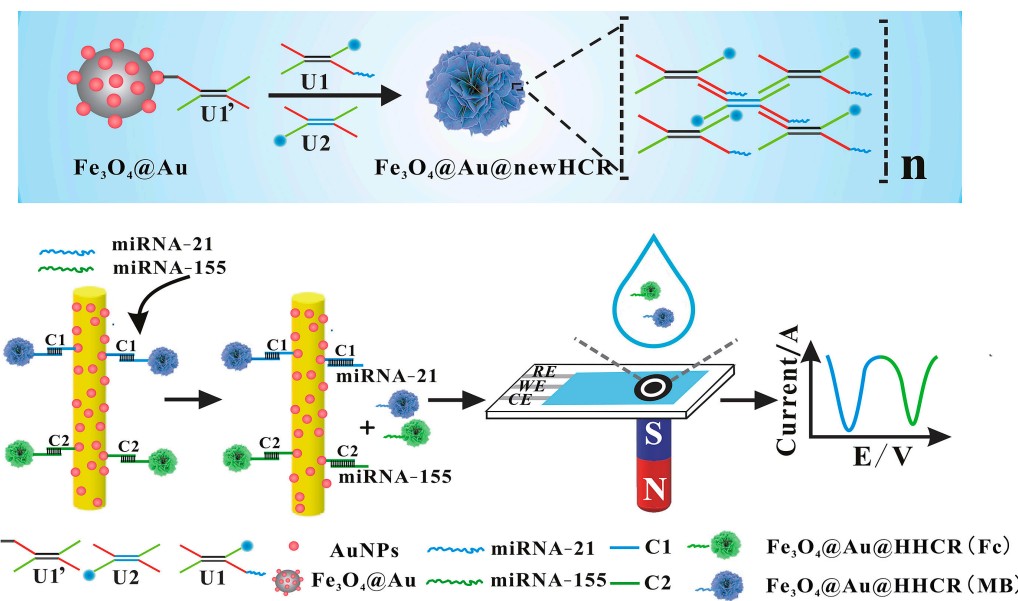

**Figure 5.** MNP-based electrochemical biosensor platform for miRNA-21 and miRNA-155. [Reproduced with the permission of [197] Copyright 2019 Published by Elsevier B.V.].

## 4. Conclusions and Future Perspective

In this review, we focus on the use of MNPs for the detection of miRNAs in biological samples. Thanks to their many advantages, magnetic nanoparticles are widely used in fields such as food, environmental, biomedical, and clinical research. In recent years, the research and development of magnetic nanoparticles have increased. MNPs are among the most highlighted nanoparticles in a variety of biomedical applications for screening, diagnosis, monitoring, and treatment of serious diseases, such as cancer. In particular, they have been used in drug release studies, therapeutic applications, and as contrast agents for imaging. The main challenge in these studies is the toxicity of MNPs, which has not been fully understood. To avoid potential toxicity, MNPs are coated and functionalized. Sometimes, the size of MNPs can also be a challenge. The success of promising results from in vitro and in vivo studies is not fully transferred to clinical applications. Similarly, the

size and shape of MNPs have major implications for their productivity. There are numerous approaches to improve the efficacy of MNPs. These include smaller sizes and biocompatible polymer or metal sheaths to improve blood circulation and shorten the time required to reach the target tissue. Basically, the solution of all MNP-related problems requires the development of appropriate synthesis procedures and the study of the obtained materials to determine the optimal behavior. From a clinical perspective, the dark spots regarding their safe use will not be resolved until the problems metabolizing from the body and toxicity are resolved.

Despite these limitations, MNPs are preferred in biosensing systems because they can separate the analyte from complex samples (urine and blood) and pre-concentrate it on the electrode surface, increasing signal and selectivity. They are very important to the field of life sciences, especially in separating complex biological samples by using the magnetic field to bind and then release the samples. The most critical problem in biosensing systems using MNPs is also finding the most suitable size, shape, and coating for surface modification. Coating magnetic nanoparticles makes it possible to produce biosensing systems targeting cancer cells or cancer-associated biomolecules. One of the cancer-associated biomolecules is miRNAs. miRNAs can act as oncogenes or tumor suppressors under certain conditions. Deregulated miRNAs have been shown to affect hallmarks of cancer, including maintenance of proliferative signaling, evasion of growth suppressors, resistance to cell death, activation of invasion and metastasis, and induction of angiogenesis. A growing number of studies have identified miRNAs as potential biomarkers for human cancer diagnosis and prognosis, as well as therapeutic targets or tools that require further investigation and validation. Therefore, extracellular/circulating miRNAs can be used for disease detection. The development of detection systems using magnetic nanoparticles is an essential area of research for the sensitive and specific detection of miRNAs present at low levels in biological samples and have stability issues. The use of magnetic nanoparticles in systems to discriminate the amount and presence of these exogenous miRNAs aims to concentrate the analyte by specific surface conjugation. Even at very low levels, miRNA can be specifically and sensitively detected. MNPs are currently used in electrochemical or optical sensor systems for the detection of miRNA, which has already been developed. However, these systems need to be further developed considering the complex structures of the samples used and the conditions affecting the measurement, such as pH, viscosity, and matrix effect. It is clear that in the future, the existing systems will be combined with machine learning and used not only for actual diseases but also for predicting diseases in screening studies.

**Supplementary Materials:** The following supporting information can be downloaded at: https://www.mdpi.com/article/10.3390/magnetochemistry9010023/s1, Table S1: Literature survey of the MNP-based biosensor system for miRNA detection.

**Author Contributions:** Conceptualization, S.B.H. and D.H.; writing—original draft preparation, S.B.H., D.H., N.U., S.E. and S.T.; writing—review and editing, S.B.H., D.H., S.E. and S.T.; supervision, S.E. and S.T. All authors have read and agreed to the published version of the manuscript.

**Funding:** This research was funded by the Republic of Turkey, Ministry of Development, grantnumber 2016K121190.

**Institutional Review Board Statement:** Not applicable.

**Informed Consent Statement:** Not applicable.

**Data Availability Statement:** Not applicable.

**Conflicts of Interest:** The authors declare no conflict of interest.

## Abbreviations

| | |
|---|---|
| MNPs | magnetic nanoparticles |
| miRNA | microRNA |
| qRT-PCR | quantitative real-time polymerase chain reaction |
| PEG | polyethylene glycol |
| PVA | polyvinyl alcohol |
| PAA | polyacrylic acid |
| TEM | transmission electron microscopy |
| SEM | scanning electron microscopy |
| DLS | dynamic light scattering |
| XRF | X-ray fluorescence |
| XRD | X-ray diffraction |
| ICP-MS | inductively coupled plasma mass spectroscopy |
| AAS | atomic absorption spectrophotometry |
| XPS | X-ray photoelectron spectroscopy |
| FT-IR | Fourier transform infrared spectroscopy |
| XAS | X-ray absorption spectroscopy |
| TGA | thermogravimetric analysis |
| RS | Raman spectroscopy |
| VSM | magnetometer |
| SQUID | superconducting quantum interference device magnetometry |
| pri-miRNAs | primary transcript |
| pre-miRNA | precursor-miRNA |
| EDC | 1-ethyl-3-(3-dimethylaminopropyl) carbodiimide hydrochloride |
| LNA | locked nucleic acid |
| cDNA | complementary DNA |
| PNA | peptide nucleic acids |
| SERS | surface-enhanced Raman spectroscopy |
| ssDNA | double-stranded DNA |
| LOD | limit of detection |
| SWV | square wave voltammetry |
| CV | cyclic voltammetry |
| DPV | differential pulse voltammetry |

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
