# Peer review of "Recent Approaches in Magnetic Nanoparticle-Based Biosensors of miRNA Detection"

_magnetochemistry, doi:10.3390/magnetochemistry9010023_

Round 1

Reviewer 1 Report

the review paper manipulates how the MNPs are used in biosensing technology, especially for MiRNAs and that is interesting but some points need to be clarified in the text:

1- the role of miRNA in the detection should be declared (i.e. how miRNA helps in the detection of diseases (viral, genetic, cancers,...) and the advantages them based on the literature)

2- the resolution of Figure 2 should be modified

3- tabulate the biosensors based on MNPs and miRNA (with references)

4- elaborate more in the conclusion and future perspectives section and focus on the importance and privileges of MNPs in the detection and eventually how the miRNA help in the early detection of the diseases

Author Response

Dear Editor,

We would like to the thank the Editor and all the reviewing parties for their interest and effort in considering our manuscript. We have taken the reports provided and given our utmost to answer and complete the manuscript to reach the expected level and quality deserved for the Journal.

We believe that the current revised manuscript has seen tremendous improvement thanks to the various enquiries and suggestions of the reviewers. We hope these enhancements and our efforts in could be seen throughout the answers to reviewers’ section (below) and other materials of the manuscript.

Please find in the following a one-by-one detailed answers for each enquiry raised by the reviewers:

Reviewer #1:

Comments:

The review paper manipulates how the MNPs are used in biosensing technology, especially for MiRNAs and that is interesting but some points need to be clarified in the text.

------------------------------------------------------------------------------------------------------------

Question #1

The role of miRNA in the detection should be declared (i.e. how miRNA helps in the detection of diseases (viral, genetic, cancers,...) and the advantages them based on the literature)

Answer #1

We thanks to the reviewer for additions. We have added additional information to this comment under the heading “3.1. miRNA” and highlighted it in yellow. We think it is now more descriptive.

------------------------------------------------------------------------------------------------------------

Question #2

The resolution of Figure 2 should be modified.

Answer #2

Thanks to the reviewer for the warning. We have corrected resolution of Figure 2 and believe it is now better.

------------------------------------------------------------------------------------------------------------

Question #3

Tabulate the biosensors based on MNPs and miRNA (with references).

Answer #3

We thank for comment. However, a table on biosensor systems and miRNAs already exists as Table S1.

--------------------------------------------------------------------------------------------------------

Question #4

Elaborate more in the conclusion and future perspectives section and focus on the importance and privileges of MNPs in the detection and eventually how the miRNA help in the early detection of the diseases.

Answer #4

We thank reviewer for suggestion. We have redesigned the "Conclusion and Future Perspective" section with your suggestions. We think it is more comprehensive now.

------------------------------------------------------------------------------------------------------------

Reviewer 2 Report

In the review article titled “Recent Approaches in Magnetic Nanoparticle-based Biosensors of miRNA Detection,” the authors have presented an overview of the use of magnetic nanoparticles (MNPs) for the detection of microRNA (miRNA). The review article is very well written and organized and does a very good job of covering the recent literature in the field. As such the article would be very useful for new researchers in the field from diverse technical backgrounds. I recommend the acceptance of this article with minor revisions as noted below.

1. Section 3.3 can be renamed as MNP-based biosensors for miRNA detection

2. Although the authors briefly describe the future perspective, the factors need to be improved for further development, and it would be illuminating to elaborate on these parameters in detail.

3. A brief discussion on the status of MNPs being employed in actual clinical settings would be helpful.

4. The authors correctly indicated a significant role of machine learning in taking the field forward, but they didn’t discuss this in the review. A brief section discussing recent literature employing machine learning and MNPs for diagnostics purposes would be good.

Author Response

Response to reviewers’ comments

Dear Editor,

We would like to the thank the Editor and all the reviewing parties for their interest and effort in considering our manuscript. We have taken the reports provided and given our utmost to answer and complete the manuscript to reach the expected level and quality deserved for the Journal.

We believe that the current revised manuscript has seen tremendous improvement thanks to the various enquiries and suggestions of the reviewers. We hope these enhancements and our efforts in could be seen throughout the answers to reviewers’ section (below) and other materials of the manuscript.

Please find in the following a one-by-one detailed answers for each enquiry raised by the reviewers:

Reviewer #2:

In the review article titled “Recent Approaches in Magnetic Nanoparticle-based Biosensors of miRNA Detection,” the authors have presented an overview of the use of magnetic nanoparticles (MNPs) for the detection of microRNA (miRNA). The review article is very well written and organized and does a very good job of covering the recent literature in the field. As such the article would be very useful for new researchers in the field from diverse technical backgrounds. I recommend the acceptance of this article with minor revisions as noted below.

------------------------------------------------------------------------------------------------------------

Question #1

Section 3.3 can be renamed as MNP-based biosensors for miRNA detection

Answer #1

We thank the reviewer. We made the change as you suggested.

------------------------------------------------------------------------------------------------------------

Question #2

Although the authors briefly describe the future perspective, the factors need to be improved for further development, and it would be illuminating to elaborate on these parameters in detail.

Answer #2

We thank the reviewer for comment. We revised the title of Conclusion and future perspective in line with the comments of the reviewers. We believe it is stronger now.

------------------------------------------------------------------------------------------------------------

Question #3

A brief discussion on the status of MNPs being employed in actual clinical settings would be helpful.

Answer #3

We thank the reviewer very much for their valuable contribution. We have included a brief discussion of the use of MNPs in real clinical settings in the conclusion and future perspective. The relevant section reads as;

“MNPs are among the most highlighted nanoparticles in a variety of biomedical applications for screening, diagnosis, monitoring, and treatment of serious diseases, such as cancer. In particular, they have been used in drug release studies, in therapeutic applications, and as contrast agents for imaging. The main challenge in these studies is the toxicity of MNPs, which has not been fully understood. To avoid potential toxicity, MNPs are coated and functionalized. Sometimes, the size of MNPs can also be a challenge. The success of promising results from in vitro and in vivo studies is not fully transferred to clinical applications. Similarly, the size and shape of MNPs have major implications for their productivity. There are numerous approaches to improve the efficacy of MNPs. These include smaller sizes and biocompatible polymer or metal sheaths to improve blood circulation and shorten the time required to reach the target tissue. Basically, the solution of all MNP-related problems requires the development of appropriate synthesis procedures and the study of the obtained materials to determine the optimal behavior. From a clinical perspective, the dark spots regarding their safe use will not be resolved until the problems metabolizing from the body and toxicity are resolved.”

------------------------------------------------------------------------------------------------------------

Question #4

The authors correctly indicated a significant role of machine learning in taking the field forward, but they didn’t discuss this in the review. A brief section discussing recent literature employing machine learning and MNPs for diagnostics purposes would be good.

Answer #4

We thank reivewer for the valuable comment. The machine learning part is not discussed in detail because the main purpose of the review is to describe MNP-based biosensing systems for miRNA detection in detail and the available literature on machine learning is limited. For these reasons, only estimates were made in the discussion section.

------------------------------------------------------------------------------------------------------------

Reviewer 3 Report

The present review manuscript entitled “Recent Approaches in Magnetic Nanoparticle-based Biosensors of miRNA Detection” authored by Simge Balaban Hanoglu describes the use of magnetic nanoparticles (MNPs) for the detection of circulating biomarkers such as microRNA (miRNA) has become trend owing to their properties, such as biocompatibility, ease of modification, high chemical stability, and mass transfer. The authors report an interesting review manuscript approach and the presentation of the work is clear. The objective and justification of the work are clear. I would recommend it for publication in Magnetochemistry. However, certain Major issues are detailed below to improve the quality of the manuscript.

I advise the authors to take the following points into account while revising their manuscript.

Comment 1: Add the list of acronyms/ abbreviations in the revised manuscript.

Comment 2: Include the Table of content in the manuscript.

Comment 3: There are some typographical errors in the manuscript text, so the authors need to correct them in the revised manuscript and also improve the English language of the manuscript.

Comment 4: Abstract should discuss the problem statement in the manuscript text. So, revise the abstract section.

Comment 5: Include the Formation mechanism of Magnetic nanoparticles in the revised manuscript.

Comment 6: Include and discuss the factors affecting Magnetic nanoparticle synthesis.

Comment 7: A more detailed and critical literature review discussion is required for all the sub-sections with their detailed pros and cons.

Comment 8: In MNP synthesis, give different approaches by doing comparative studies with the recent literature. This can be compared by giving Tables as well as Figures.  

Comment 9:  Please check whether the authors have already obtained the copyright permissions for Figure 1 used in the manuscript.

Comment 10: Characterization of Magnetic nanoparticles is weak. Comparative and extensive discussion is needed with Figures and Tables of each characterization technique. 

Comment 11: The conclusion section should be elaborated.

Comment 12: The homogeneity of the reference section needs to be maintained. In some references, journal names are written in full form and some in abbreviation form. So please check and revise accordingly to the journal instructions.

Author Response

Response to reviewers’ comments

Dear Editor,

We would like to the thank the Editor and all the reviewing parties for their interest and effort in considering our manuscript. We have taken the reports provided and given our utmost to answer and complete the manuscript to reach the expected level and quality deserved for the Journal.

We believe that the current revised manuscript has seen tremendous improvement thanks to the various enquiries and suggestions of the reviewers. We hope these enhancements and our efforts in could be seen throughout the answers to reviewers’ section (below) and other materials of the manuscript.

Please find in the following a one-by-one detailed answers for each enquiry raised by the reviewers:

------------------------------------------------------------------------------------------------------------

Reviewer #3:

The present review manuscript entitled “Recent Approaches in Magnetic Nanoparticle-based Biosensors of miRNA Detection” authored by Simge Balaban Hanoglu describes the use of magnetic nanoparticles (MNPs) for the detection of circulating biomarkers such as microRNA (miRNA) has become trend owing to their properties, such as biocompatibility, ease of modification, high chemical stability, and mass transfer. The authors report an interesting review manuscript approach, and the presentation of the work is clear. The objective and justification of the work are clear. I would recommend it for publication in Magnetochemistry. However, certain Major issues are detailed below to improve the quality of the manuscript.

------------------------------------------------------------------------------------------------------------

Question #1

Add the list of acronyms/ abbreviations in the revised manuscript.

Answer #1

We added a list of abbreviations before introduction.

------------------------------------------------------------------------------------------------------------

Question #2

Include the Table of content in the manuscript.

Answer #2

We added a list of abbreviations after abstract part.

------------------------------------------------------------------------------------------------------------

Question #3

There are some typographical errors in the manuscript text, so the authors need to correct them in the revised manuscript and also improve the English language of the manuscript.

Answer #3

We would like to thank the reviewer. All necessary grammer, typo revision were done. We have also checked it with the Grammarly program. We have uploaded the received report into the system.

------------------------------------------------------------------------------------------------------------

Question #4

Abstract should discuss the problem statement in the manuscript text. So, revise the abstract section.

Answer #4

The abstract has been revised.

Question #5

Include the Formation mechanism of Magnetic nanoparticles in the revised manuscript.

Answer #5

We thank the reviewer for comment. Under the "2.1. Synthesis" part, we have added sample reactions describing the formation mechanisms for the synthesis methods.

------------------------------------------------------------------------------------------------------------

Question #6

Include and discuss the factors affecting Magnetic nanoparticle synthesis.

Answer #6

We have given the factors affecting synthesis as advantages and disadvantages for each synthesis method. In addition, we have tried to illustrate them with some studies from recent years. Now we believe that the part "2.1.Synthesis" is stronger.

------------------------------------------------------------------------------------------------------------

Question #7

A more detailed and critical literature review discussion is required for all the sub-sections with their detailed pros and cons.

Answer #7

The entire article has been re-edited with your valuable contributions and comments.

----------------------------------------------------------------------------------------------------------

Question #8

In MNP synthesis, give different approaches by doing comparative studies with the recent literature. This can be compared by giving Tables as well as Figures.  

Answer #8

We thank the reviewer for suggestion. We have included a table (Table 1) with examples from recent publications for each synthesis method for your reference. We have also discussed the advantages and disadvantages of these publications in the text.

----------------------------------------------------------------------------------------------------------

Question #9

Please check whether the authors have already obtained the copyright permissions for Figure 1 used in the manuscript.

Answer #9

Thanks to the reviewer for the warning. However, Figure 1 was created by us and does not require copyright.

----------------------------------------------------------------------------------------------------------

Question #10

Characterization of Magnetic nanoparticles is weak. Comparative and extensive discussion is needed with Figures and Tables of each characterization technique. 

Answer #10

We thank the reviewer for comment. We have expanded the section "2.3. Characterization" and included a figure summarizing the methods we reviewed as Figure 1. We believe the chapter is now more informative.

----------------------------------------------------------------------------------------------------------

Question #11

The conclusion section should be elaborated.

Answer #11

We thank reviewer for suggestion. We have redesigned the "Conclusion and Future Perspective" section with your suggestions. We think it is more comprehensive now.

---------------------------------------------------------------------------------------------------------

Question #12

The homogeneity of the reference section needs to be maintained. In some references, journal names are written in full form and some in abbreviation form. So please check and revise accordingly to the journal instructions.

Answer #12

We apologize for the confusion. Now we have fixed them all.

---------------------------------------------------------------------------------------------------------

Round 2

Reviewer 1 Report

the review article is ready for publication

Reviewer 3 Report

The authors are revised their manuscript according to my comments. So the manuscript should be accepted for publication in its current form.